# Evaluation and Monitoring of the Natural Toxin Ptaquiloside in Bracken Fern, Meat, and Dairy Products

**DOI:** 10.3390/toxins15030231

**Published:** 2023-03-21

**Authors:** Hana Park, Yoeseph Cho, JiEun Lee, Kang Mi Lee, Ho Jun Kim, Jaeick Lee, Yong-Sun Bahn, Junghyun Son

**Affiliations:** 1Doping Control Center, Korea Institute of Science and Technology, Seoul 02792, Republic of Korea; 2Department of Biotechnology, College of Life Science and Biotechnology, Yonsei University, Seoul 03722, Republic of Korea; 3Department of Microbiology and Immunology, Institute for Immunology and Immunological Diseases, Brain Korea 21 PLUS Project for Medical Science, Yonsei University College of Medicine, Seoul 03722, Republic of Korea; 4KnA Consulting, Yongin-si 16942, Republic of Korea

**Keywords:** ptaquiloside, method validation, LC–MS/MS, QuEChERS (quick, easy, cheap, effective, rugged and safe), bracken fern, monitoring, exposure assessment

## Abstract

Ptaquiloside, a naturally occurring cancer-causing substance in bracken fern, has been detected in the meat and milk of cows fed a diet containing bracken fern. A rapid and sensitive method for the quantitative analysis of ptaquiloside in bracken fern, meat, and dairy products was developed using the QuEChERS method and liquid chromatography–tandem mass spectrometry. The method was validated according to the Association of Official Analytical Chemists guidelines and met the criteria. A single matrix-matched calibration method with bracken fern has been proposed, which is a novel strategy that uses one calibration for multiple matrices. The calibration curve ranged from 0.1 to 50 µg/kg and showed good linearity (*r*^2^ > 0.99). The limits of detection and quantification were 0.03 and 0.09 µg/kg, respectively. The intraday and interday accuracies were 83.5–98.5%, and the precision was <9.0%. This method was used for the monitoring and exposure assessment of ptaquiloside in all routes of exposure. A total of 0.1 µg/kg of ptaquiloside was detected in free-range beef, and the daily dietary exposure of South Koreans to ptaquiloside was estimated at up to 3.0 × 10^−5^ µg/kg b.w./day. The significance of this study is to evaluate commercially available products in which ptaquiloside may be present, to monitor consumer safety.

## 1. Introduction

Bracken fern (*Pteridium aquilinum*) is among the five most abundant plant species in the world [1] and is known to cause cancer in animals [2]. Ptaquiloside (PTA) is the most poisonous toxin in bracken fern and exhibits the following biological properties in livestock and laboratory animals: acute bracken fern poisoning; bright blindness; and mutagenic, mitogenic, and carcinogenic effects [2,3,4]. Bracken fern may also increase the risk of carcinogenesis in humans. Several studies have found a correlation between the ingestion of bracken fern and esophageal and stomach cancer [5,6,7,8,9].

Nevertheless, bracken fern is used worldwide in food, traditional medicine, and food supplements. In Korea, Japan, China and Brazil, traditional food containing bracken fern (*P. aquilinum*) is very popular and young crosiers (46 unfolded young fronds; 15–40 cm; *P. aquilinum*) are intentionally consumed [10,11].

In addition to consuming bracken fern, consuming foods such as meat, milk and dairy products may be potential sources of human exposure to PTA. When edible plants are limited, animals may graze on polluted pastures or consume hazardous plants. This may have an effect on animal health and result in indirect human exposure via contaminated animal products [12]. PTA has also been detected in the milk of cows provided a known amount of bracken fern [13,14], as well as in the milk of animals that grazed on bracken fern [15,16].

The analysis of PTA is challenging because of its unstable nature. PTA can be transformed into *pterosin* B (PTB), which is considered nontoxic, via hydrolysis, heat, and the presence of enzymes under acidic or alkaline conditions. Therefore, PTA quantification methods comprise measuring PTA and PTB simultaneously, [17,18,19,20] or measuring it after the total conversion of PTA into PTB [2,21,22]. The PTA content of samples cannot be measured accurately because of the insufficient hydrolysis of PTA into PTB. The conversion method has the disadvantage of not distinguishing between *pterosins* initially present in the analyte sample and *pterosins* intentionally generated during sample treatment. Therefore, a specific, accurate and rapid analytical approach is required to quantify PTA.

A sensitive and standardized sample preparation process is required to maintain sample quality and stability. PTA slowly degrades in the pH range of 4.4–6.4 and at low temperatures [23]. The QuEChERS method-based European Committee for Standardization (CEN) standard EN 15662 involves the addition of four salts, namely magnesium sulfate (MgSO_4_), sodium chloride (NaCl), sodium citrate (Na_3_C_6_H_5_O_7_), and sodium hydrogen citrate sesquihydrate (C_6_H_6_Na_2_O_7_·1.5H_2_O), to a citrate buffer at a weight ratio of 4:1:1:0.5 (*w*/*w*). The addition of salts to the buffer allows a consistent pH of 5–5.5 to be maintained during the extraction procedure, which usually provides a recovery value of ≥70% for compounds that are sensitive to bases or acids. Hence, the QuEChERS method is considered as an effective approach to prevent the decomposition of PTA during the sample preparation process.

Based on the significance of PTA for human safety, many studies have been conducted to quantify it using LC-MS analysis, which is highly sensitive and selective for the detection of low levels of natural toxins in complex matrices [24]. These studies have been performed to quantify PTA in various matrices, including biological fluids such as plasma and urine, environmental samples such as ground water and food matrices such as milk, highlighting the versatility of LC-MS analysis [17,18,19,20,25,26,27]. However, previous quantitative methods developed for PTA analysis usually only deal with one or two matrices with similar physical properties, resulting in a limited number of samples that can be analyzed using the developed methods. Furthermore, most studies of PTA have focused on quantifying the content in wild foods, such as raw bracken fern and unpasteurized milk from farm-raised cows [28,29], so there is insufficient information on content in commercially available products. Recently, a study monitored the PTA content in commercial food products and European natural remedies obtained from online stores, including Korean bracken fern products, but the number of monitoring samples was too small, and the variety was limited to dried bracken fern, affording poor diversity [11]. Moreover, LC-MS and SPE (solid-phase extraction) pre-treatment methods have primarily been used for PTA analysis [18,19,20,25,27], and a direct detection method combining LC-MS/MS and QuEChERS for the analysis of PTA in food has not yet been developed.

This study is the first survey on determining the content of PTA in commercial bracken fern products and processed foods conducted in Korea. All the routes through which PTA enters humans, such as bracken fern, meat, and milk, were included as monitoring samples. The transfer of PTA from bracken fern to livestock and finally to humans highlights the importance of quantifying PTA levels not only in bracken fern, but also in animal-derived foods. In addition, matrix-matched calibration was performed using preprocessed bracken fern in which no PTA was detected. Our proposed method is capable of detecting and quantifying PTA in various food matrices and bracken ferns using a single matrix-matched calibration approach. To ensure food safety, it is important to establish a rapid and accurate method for quantifying the PTA content of foods. In this study, we combined QuEChERS and LC-MS/MS to develop a method that can quickly and directly quantify PTA by overcoming the limitations of conventional PTA quantification methods. The analytical method developed in this way has the ultimate purpose of the final detection and quantitative analysis of PTA in various foods, including bracken fern sold in Korea, and the evaluation of food safety through exposure assessment.

## 2. Results

### 2.1. Method Development

#### 2.1.1. Optimization of MS Conditions

A 1 mg/L quantity of a PTA standard working solution was infused to obtain a full scan acquisition and MS/MS spectrum. The fully scanned mass spectrum showed the most abundant peak at *m/z* 399, equivalent to [M + H] ^+^. Three main product ions with *m/z* 181, 277, and 381 were obtained when the protonated molecule *m/z* 399 was selected as the precursor ion (Appendix A). The product ion with the highest abundance (*m/z* 181) was chosen for quantitative purposes, while the remaining two (*m/z* 277 and 381) were chosen for qualification (Figure 1). Appendix A lists multiple reaction monitoring (MRM) parameters and the retention time for PTA. The mobile phase composition was also investigated, to select the optimal conditions for PTA determination. Appendix A summarizes the optimized MS/MS parameters for PTA determination.

#### 2.1.2. Optimization of Chromatographic Separation

Chromatographic parameters, including flow rate, mobile phase, and column temperature, were optimized to achieve the best analyte (PTA) separation in the shortest time. For this, different linear gradients of mobile phase A (water with 0.1% of formic acid) and mobile phase B (acetonitrile with 0.1% of formic acid), column temperatures (35 °C, 40 °C, and 50 °C) and flow rates (0.3, 0.5, 0.6, and 0.7 mL/min) were compared. The total run time was 5 min. We optimized the LC-MS/MS parameters for detecting PTA in foods by balancing sensitivity and resolution, while minimizing negative effects due to its heat sensitivity. Increasing the oven temperature can improve resolution and peak shape, but this slightly decreased the sensitivity due to the analyte’s nature. Additionally, a flow rate that is too low reduces sensitivity, while a flow rate that is too high broadens the peak shape. We selected optimal parameters of a 35 °C oven temperature and a 0.5 mL/min flow rate based on the chemical properties of PTA, the instrument performance, and the previous literature. An analysis time of 5 min was chosen for practical and efficient detection without compromising resolution or sensitivity.

### 2.2. Method Validation

A method-validation study was conducted to evaluate the selectivity, linearity, limit of detection (LOD), limit of quantification (LOQ), accuracy, precision, and recovery of the developed method for PTA.

#### 2.2.1. Selectivity

Method selectivity can be assessed by comparing the matrix (without the target analyte) to the matrix spiked with the target analyte [30].

Figure 2 shows MRM chromatograms of the blank matrix (Figure 2A), standard working solution (0.1 µg/mL of PTA, Figure 2B), sample spiked with PTA on blank matrix (100 µg/kg, Figure 2C), and a real sample (sample ID Z; see Table 2 and Figure 2D). The blank matrix refers to a bracken fern sample in which PTA is not detected. We established that there were no other significant interfering peaks near the chromatographic peak (retention time = 3.21 min). The matrix of the sample did not affect tandem mass spectrometry. The MRM chromatogram of the real sample (Figure 2D) showed excellent, sharp peak shapes. 

#### 2.2.2. Linearity and LOD and LOQ

A calibration curve was constructed via matrix-matched standard calibration using the processed fern matrix, in which PTA was not detected at concentrations of 0.1, 1, 2, 10, 20 or 50 µg/kg, and the linearity (*r*^2^) of the calibration curve was 0.9979. Good linearity was observed, with coefficients of determination (*r*^2^) of >0.99. LOD and LOQ were determined based on the slope and standard deviation (σ) of the linear coefficient of the analytical curve. The LOD and LOQ values were 0.03 and 0.09 µg/kg, respectively. These results indicate that the sensitivity of the proposed method is suitably high (Appendix A).

#### 2.2.3. Accuracy and Precision

Accuracy can be expressed as the proximity of a result or the average of a series of measurements to the true value. This was established by evaluating blank samples spiked with PTA standard working solutions at three concentrations (0.1, 20, and 50 g/kg) in triplicate, and by comparing the concentrations of the extracted PTA with those from the MMS (matrix-matched standard) calibration curves. The results are listed in Table 1. The intraday and interday accuracies ranged from 83.5 to 98.5% and 83.8 to 97.5%, respectively. Precision was estimated using relative standard deviation (RSD), and intraday and intraday precisions ranged from 0.4 to 4.9% and 1.5 to 7.2%, respectively. The developed method afforded satisfactory accuracy and excellent precision. The assessed accuracy and precision were acceptable in accordance with Official Methods of Analysis Guidelines (2016) for Standard Method Performance Requirements of the Association of Official Agricultural Chemists (AOAC) [31].

#### 2.2.4. Comparison of the Relative Differences of the Same Concentration Analyte in Various Matrices

To confirm that the developed method can be applied to various matrices, such as bracken fern, meat and milk, the same concentration of PTA was spiked into each matrix to test the difference in the area value for PTA. Briefly, 20 µg/kg (medium concentration) of PTA was spiked into bracken fern, meat and milk matrices, and the area value difference (%) with the PTA standard working solution (20 µg/kg) was compared. As a result of comparing the average values of three repeated experiments, the area value decreased by 4.0% in bracken fern, decreased by 4.1% in meat, and increased by 5.3% in milk compared to the area value for 20 µg/kg of the PTA standard working solution. There was no significant difference between the peak area value of the PTA standard working solution and the peak area value of PTA added to each matrix (Appendix A). Furthermore, the differences in the slopes of the matrix-matched calibration curves for each matrix, as well as the PTA standard working solution calibration curve, were compared. Selected concentrations of PTA were added to each matrix (bracken fern, meat, and milk) and the relative differences with PTA standard working solutions were compared to determine the matrix effects. This experiment was conducted solely for the purpose of comparing relative differences under the same conditions and was separate from obtaining the calibration curve used to calculate the concentration. The slopes of the PTA standard calibration curve and each matrix-matched standard calibration curve did not differ significantly between 2.3% and 4.1%. Furthermore, the coefficient of variation (CV, %) value of each slope was 1.7%. This confirmed the similarity between the samples and the standard working solution. The variability of the slope of the standard line (the precision of the slope of the standard line expressed as a CV (%)) can be used as a useful predictor of the relative matrix effects. This precision value should not exceed 3–4% to ensure the method’s reliability and absence of relative matrix effects. Hence, the developed method has no matrix effect and is applicable to various matrices, such as bracken fern, meat, and milk (Appendix A).

#### 2.2.5. Extraction Recovery Rate

The recovery rate (RE) of the extraction procedure was evaluated at three concentration levels (0.1, 20, and 50 µg/kg) of the two peak area ratios for the standard working solution (PTA), spiked before and after pretreatment of the bracken fern samples. The results showed excellent recovery rates of 71.2–93.6% at each concentration when assayed in quadruplicate, demonstrating consistency and reproducibility (Appendix A).

#### 2.2.6. Cross-Validation

The assay was cross-validated by two institutions (Organizations A and B). In both cases, Vanquish UHPLC–TSQ Altis (Thermo Fisher Scientific, Waltham, MA, USA) was used as the analytical instrument and the same concentration of quality controls (0.1, 20, and 50 μg/kg) and analytical conditions were employed for cross-validation. Organization A exhibited an accuracy of 94.1–113.7% and a precision of 1.1–5.2%. Organization B showed an accuracy of 88.3–101.0% and a precision of 1.0–2.8%. Both institutions confirmed that the accuracy and precision satisfied the standards of the AOAC guidelines [31].

### 2.3. Application and Monitoring of Real Samples

The validated method was applied to analyze real commercial products, to evaluate its applicability. Foods with a history of or a high probability of PTA being detected were monitored, and 26 commercial samples were collected from local Korean markets and online markets. Among the 26 samples, 18 were bracken fern. These 18 bracken fern samples comprised seven types of dried bracken fern (Sample IDs: A–G), four types of boiled bracken fern (Sample IDs: H–K), and seven types of bracken processed food (Sample IDs: L–R). In addition, five free-range dairy products and three free-range beef products were monitored. In bracken fern (sample ID: B) the PTA detected was below the LOQ, and 0.1 µg/kg of PTA was detected in free-range beef (sample ID: Z) (Table 2).

### 2.4. Assessment of Dietary Exposure to PTA

The dietary exposure to PTA was assessed herein. Food consumption and average weight were determined using the KNHANES (National Health and Nutrition Examination Survey) data [32]. Table 2 lists the PTA exposure–estimation values based on the PTA contents in vegetables, meat, and dairy products. The estimated exposure to PTA from food consumption varied from 0.0 to 3.0 × 10^−5^ g/kg b.w./day using the lower bound (LB) and upper bound (UB) approaches (0.0–1.1 × 10^−6^ µg/kg b.w./day in vegetables, 0.0–3.0 × 10^−5^ µg/kg b.w./day in dairy products and 2.5 × 10^−5^–2.5 × 10^−5^ µg/kg b.w./day in meat). Although PTA was detected only in the meat sample among the food products analyzed in this study, vegetables and dairy products showed dietary exposure to PTA because of the large frequency rate, such as the vegetable as it is, and the milk used in this study. The upper concentration limit of each exposure was determined based on the dietary intake data. For bracken fern, the intake rate of dried, blanched, boiled, and processed forms may vary. Similarly, for milk and yogurt products, the upper concentration limit may vary depending on whether they were organic or pastured, and on the specific cut of beef used. Therefore, the samples with the same N.D (not detected) results had different upper bound values due to these factors.

## 3. Conclusions

PTA is a carcinogen found in bracken fern that can be transferred to humans through livestock via meat and milk; therefore, quantifying this substance from the viewpoint of food safety is important. However, no method has been established to quantify PTA rapidly and accurately in various food matrices. In this study, we developed a rapid, sensitive, and simple method for the quantitative analysis of PTA using LC–MS/MS, following an efficient QuEChERS cleanup protocol. The method developed in this way can be useful for monitoring PTA levels in foods and ensuring that they are safe for human consumption. The validated method was applied to 26 commercial products. For the first time, commercially available bracken fern, meat, and dairy products were subjected to tests for detecting their level of PTA. Thus, all possible sources of PTA contamination in the entire food chain can be monitored. We confirmed through several tests that the proposed method is promising for determining the concentration of PTA present in various food matrices. As a result, PTA below the LOQ was detected in only one commercially available bracken fern product and was not detected in the other products. PTA exists in raw bracken fern, while most of the bracken sold undergoes pre-treatment such as blanching or drying after boiling, to remove or reduce its concentration. However, sellers or manufacturers do not usually provide detailed information about their own pretreatment, making it difficult to compare differences between samples. For sample B, PTA was detected below LOQ, indicating that the pretreatment may have been insufficient. In addition, 0.1 µg/kg of PTA was detected in free-range beef obtained from New Zealand. Sample Z (beef), also obtained from the market, had only limited information; that it was sourced from 100% grass-fed and free-range cattle in New Zealand. It is possible that small amounts of PTA were introduced randomly to cattle through grazing. Although the amount of PTA detected in beef is small, it is difficult to say that it is safe because PTA can accumulate in the body and cause cancer. Fortunately, PTA was not detected in most of the samples. However, this study confirmed the presence of PTA in some randomly selected samples. Therefore, it is important to develop appropriate food safety measures to ensure consumer safety. The PTA analysis method developed herein is expected to be used as a test method for the safety management of food toxins in the future. Future studies would monitor more meat and dairy products. Toxicity studies on PTA conducted thus far are insufficient. Therefore, only dietary exposure was evaluated in this study, and risk assessment was excluded. Through this study, we would like to emphasize that various and continuous food monitoring is necessary for the food safety management of PTA, and that toxicity studies based on food exposure assessment should, additionally, be performed.

## 4. Materials and Methods

### 4.1. Chemicals and Reagents

All chemicals and solvents used were of HPLC or analytical grade. PTA (purity > 98%, CAS No. 87625-62-5) was purchased from Chemfaces (Wuhan, Hubei, China) and dimethyl sulfoxide (glacial ≥ 99.7% HPLC grade, CAS No. 67-68-5) was supplied by Sigma-Aldrich (St. Louis, MO, USA). Acetonitrile (glacial ≥ 99.9% HPLC grade, CAS 75-05-8) was purchased from J.T. Baker (Phillipsburg, NJ, USA). Water with 0.1% of formic acid (Optima LC–MS Grade) and acetonitrile with 0.1% of formic acid (Optima LC–MS grade) were acquired from Fisher Scientific (Pittsburgh, PA, USA). The solvent, i.e., 0.1% formic acid in water (LC–MS Grade), used for the sample dilution before analysis was purchased from Thermo Fisher Scientific (Waltham, MA, USA). Deionized water (18.2 MΩ cm) was obtained from a Milli-Q pure water system (MilliQ EQ 7000, Merck–Millipore, Darmstadt, Germany). The QuEChERS EN extraction packets and QuEChERS dispersive solid-phase extraction kit for fatty matrices and highly pigmented matrices were acquired from Agilent Technologies (Santa Clara, CA, USA).

### 4.2. Preparation of Standard Solutions

A 1 g/L standard stock solution of PTA in dimethyl sulfoxide was prepared. The standard working solutions were made by serially diluting standard stock solutions with acetonitrile to provide concentrations of 100, 10, 1 and 0.1 mg/L. PTA has poor thermal stability. To address this, we prepared all standard solutions in amber tubes and stored them in a freezer at −20 °C until use to maximize stability. Additionally, to minimize potential degradation, we freshly prepared standard working solutions and used them within 2 weeks.

### 4.3. Sample Preparation

Bracken fern samples were collected from online markets. For extraction, 10 g of homogenized sample was weighed into a 50 mL centrifuge tube. Samples were frozen prior to extraction to prevent loss of recovery due to heat caused by magnesium sulfate (MgSO_4_). And then 10 mL of acetonitrile was used as extraction solvent and the samples were agitated vigorously for 30 s with Vortex-Genie 2T (Scientific Industries, Bohemia, NY, USA) and shaken for 1 min. The QuEChERS EN (European) extraction kit, including 4 g of magnesium sulfate (MgSO_4_), 1 g of sodium chloride (NaCl), 1 g of sodium citrate (Na_3_C_6_H_5_O_7_) and 4 g sodium hydrogen citrate sesquihydrate (C_6_H_6_Na_2_O_7_·1.5H_2_O), was then added and the whole vigorously shaken for 10 min. Subsequently, the tube was centrifuged at 4000× *g* for 10 min at 4 °C. After the analyte was extracted, d-SPE (dispersive solid-phase extraction) was used for purification. The supernatant (1 mL) was transferred to a 2 mL centrifuge tube containing sorbents for solid-phase extraction. Two clean-up sorbent kits were selectively used for each matrix based on the two characteristics of the samples: pigment-dominant and fat-dominant. Module 1 utilized a dispersive-SPE kit designed for fruits and vegetables with high pigment (PN: 5982-5221), containing 25 mg of primary secondary amine (PSA), 2.5 mg of graphitized carbon black (GCB), and 150 mg of magnesium sulfate (MgSO_4_). Module 2, on the other hand, used a dispersive-SPE kit designed for fruits and vegetables with fats and waxes (PN: 5982-5121), containing 25 mg of primary secondary amine (PSA), 25 mg of end-capped octadecylsilane (C18EC), and 150 mg of magnesium sulfate (MgSO_4_). The tube was agitated for 1 min before being centrifuged at 12,700× *g* for 10 min at 4 °C. Before the LC–MS/MS analysis, the upper organic layer was diluted 10 times with 0.1% formic acid in water and transferred to an injection vial (Figure 3).

### 4.4. LC–MS/MS Conditions

The LC–MS/MS system comprised a Vanquish UHPLC–TSQ Altis mass spectrometer (Thermo Fisher Scientific, Waltham, MA, USA). Chromatographic separation was performed on a Kinetex 2.6-μm C18 column (100 mm × 2.1 mm). The mobile phase comprised A (water with 0.1% of formic acid) and B (acetonitrile with 0.1% of formic acid). Each 10 μL sample was injected and separated using a gradient elution for 0–0.5 min with 5% B, 0.5–3 min with 5–95% B, 3–4 min with 95% B, 4–4.1 min with 95–5% B and 4.1–5 min 5% B at a flow rate of 0.5 mL/min. The total chromatographic run time was 5 min. The detection of PTA was performed in MRM modes using electrospray positive ionization. The temperature of the column was maintained at 35 °C. For the mass spectrometric analysis, the parameters of the ion source were set as follows: ion spray voltage = 3500 V, sheath gas = 50 Arb, aux gas = 10 Arb, sweep gas = 1 Arb, ion transfer tube temperature = 325 °C, vaporization temperature = 350 °C and dwell time = 48 ms. Three MRM transitions were chosen for the target compound (one for quantification and the others for confirmation). The TraceFinder 4.1 software (Thermo Fisher Scientific, Waltham, MA, USA) was utilized to analyze the quantitative data collected from the calibration standards and samples.

### 4.5. Method Validation

The developed method was validated, including linearity, accuracy, precision, LOQ and LOD parameters. The matrix-matched standard was used to generate a calibration curve at six concentrations: 0.1, 1, 2, 10, 20 and 50 g/kg. The curve was created by graphing the peak area (signal intensities) as a function of the nominal calibration standard concentration. Using the least squares regression method, the linearity of the calibration curve was evaluated. The LOD and LOQ values were estimated using a method based on the response’s standard deviation and the slope of the calibration curve. The lowest concentration of PTA in the calibration curve was analyzed five times. It was calculated using Equations (1) and (2), proposed by the AOAC guideline [31], utilizing the slope of the calibration curve and the standard deviation (SD) of the five replicated analyses of the lowest concentration. The LOD and LOQ values were estimated using the following formulas: LOD = 3.3 × SD/S(1)
LOQ = 10 × SD/S(2)
where SD is the standard deviation of the y-intercept, and S represents the slope of the calibration curve. The accuracy was estimated using the following formula: mean observed concentration of spiked samples derived from matrix-matched standard/spiked concentration × 100 (%). The precision value was expressed as the coefficient of variation (CV, %) and determined from replicate analyses (n = 3). The extraction recovery was assessed by comparing the peak areas of samples spiked before and after extraction. Quantitative recovery reached 100% when the peak area of an analyte was the same for samples spiked before and after the extraction protocol.

### 4.6. Exposure Assessment of PTA

The exposure assessment of PTA was based on monitoring data, daily consumption value, and the average human body weight (57.6 kg), collected from the Korea National Health and Nutrition Survey [32]. The estimated daily consumption was calculated as follows:EDI (µg/kg b.w./day) = daily consumption (g/day) × PTA concentration (µg/kg)/average body weight (kg)(3)

Lower bound (LB) and upper bound (UB) were used in the dietary exposure estimates to provide a range of potential exposure levels. The lower bound represents the minimum possible exposure level, while the upper bound represents the maximum possible exposure level based on analytical measurements. This range can provide a more accurate estimate of potential exposure levels compared to just using a single value. In case a substance is not detected (N.D.) in a sample, a value of zero is assigned as the lower bound and the LOD is used as the upper bound. In addition, the statistics on food intake in the general population, including the average and extreme percentile (95th percentile), were also considered in the estimation of dietary exposure.

## Figures and Tables

**Figure 1 toxins-15-00231-f001:**
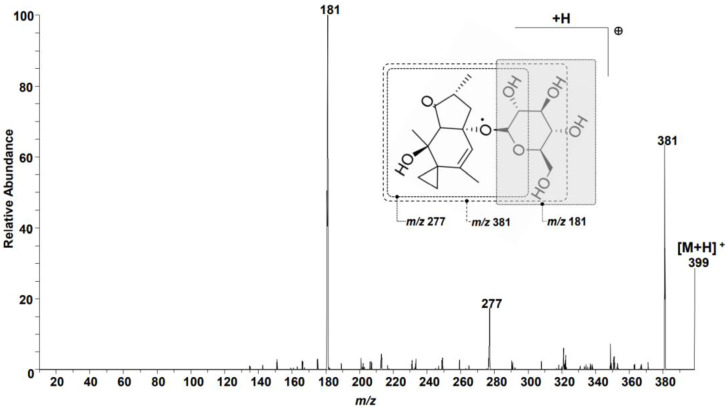
Product ion mass spectrum observed for ptaquiloside (PTA) with precursor ion *m/z* 399 in a standard working solution. Three major product ions are observed at *m/z* 181, *m/z* 277 and *m/z* 381 in this tandem mass spectrometry (MS/MS) spectrum.

**Figure 2 toxins-15-00231-f002:**
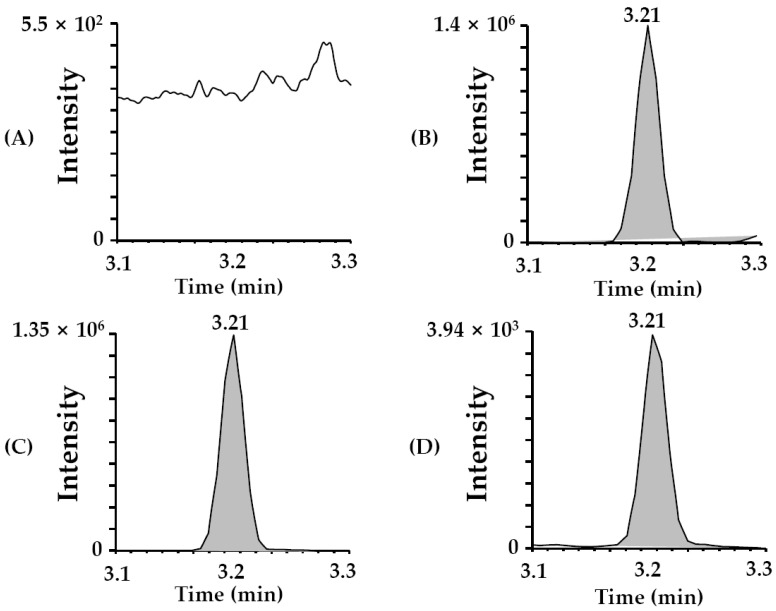
Representative chromatograms of PTA in the multiple reaction monitoring (MRM) mode for (**A**) the blank matrix, (**B**) the standard working solution (0.1 µg/mL of PTA), (**C**) the spiked sample in the blank matrix (100 µg/kg of PTA), and (**D**) the selected beef sample (sample ID: Z (see Table 2)).

**Figure 3 toxins-15-00231-f003:**
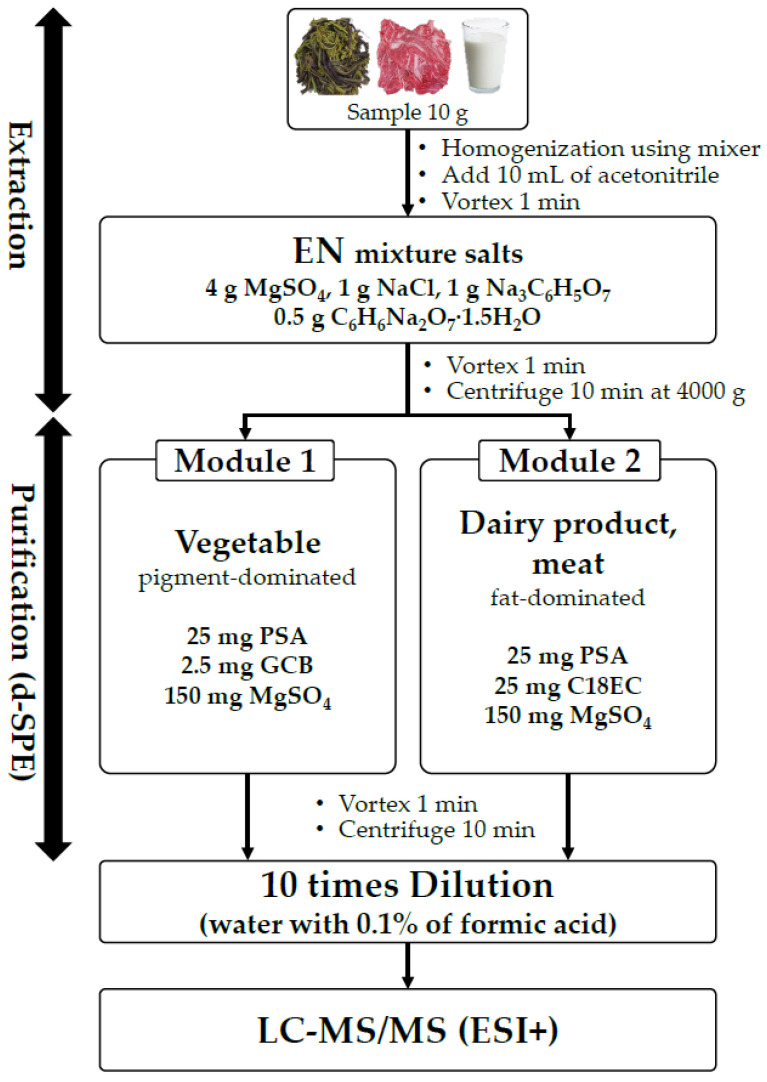
Modular extraction procedure for different characteristics of samples. EN (European), PSA (primary secondary amine), GCB (graphitized carbon black), C18EC (end-capped octadecylsilane).

**Table 1 toxins-15-00231-t001:** Validation results for the accuracy, precision, and recovery of QC samples (n = 3) at three selected levels of ptaquiloside (PTA) concentrations (0.1, 20 and 50 µg/kg).

Concentration Added (µg/kg)		Intraday (n = 3) ^a^					Interday (n = 3) ^a^					Recovery (%) ^d^
		Concentration Found (µg/kg)			Accuracy (%) ^b^	CV (%) ^c^	Concentration Found (µg/kg)			Accuracy (%) ^b^	CV (%) ^c^	
0.1	(low)	0.08	±	0.007	83.5	4.9	0.08	±	0.003	83.8	3.8	71.2
20	(medium)	19.9	±	0.08	99.5	0.4	18.8	±	1.4	94.0	7.2	93.6
50	(high)	49.3	±	0.5	98.5	1.0	48.8	±	0.8	97.5	1.5	92.6

^a^: Average of measurements conducted each day for three days within a week. ^b^: Accuracy (%) = (measured mean value/theoretical value) × 100. ^c^: Coefficient of variation (CV, %) = (the relative standard deviation/measured mean value) × 100. ^d^: Extraction recovery (%) = (pre-extraction spiked matrix/post-extraction spiked matrix) × 100.

**Table 2 toxins-15-00231-t002:** Results of monitoring the PTA content and exposure assessment.

Category ^a^	ID ^b^	Raw Materials	Modular Extraction	Area ^c^	PTA (µg/kg)	Exposure Estimation (µg/kg b.w./day)	
						Lower Bound ^j^	Upper Bound ^k^
Vegetables	A ^d^	Bracken fern	Module-1	Korea	N.D.	0.0	2.6 × 10^−7^
	B ^d^	Bracken fern	Module-1	Korea	<LOQ	0.0	6.3 × 10^−7^
	C ^d^	Bracken fern	Module-1	China	N.D.	0.0	1.1 × 10^−6^
	D ^d^	Bracken fern	Module-1	Korea	N.D.	0.0	1.1 × 10^−6^
	E ^d^	Bracken fern	Module-1	Korea	N.D.	0.0	1.1 × 10^−6^
	F ^d^	Bracken fern	Module-1	China	N.D.	0.0	1.1 × 10^−6^
	G ^d^	Bracken fern	Module-1	Korea	N.D.	0.0	1.1 × 10^−6^
	H ^e^	Bracken fern	Module-1	Korea	N.D.	0.0	1.1 × 10^−6^
	I ^e^	Bracken fern	Module-1	Korea	N.D.	0.0	1.1 × 10^−6^
	J ^e^	Bracken fern	Module-1	Korea	N.D.	0.0	6.3 × 10^−7^
	K ^e^	Bracken fern	Module-1	China	N.D.	0.0	6.3 × 10^−7^
	L ^f^	Bracken fern	Module-1	Korea	N.D.	0.0	6.3 × 10^−7^
	M ^f^	Bracken fern	Module-1	China	N.D.	0.0	2.0 × 10^−8^
	N ^f^	Bracken fern	Module-1	China	N.D.	0.0	2.0 × 10^−8^
	O ^f^	Bracken fern	Module-1	Korea	N.D.	0.0	1.1 × 10^−6^
	P ^f^	Bracken fern	Module-1	China	N.D.	0.0	1.1 × 10^−6^
	Q ^f^	Bracken fern	Module-1	China	N.D.	0.0	1.1 × 10^−6^
	R ^f^	Bracken fern	Module-1	Korea	N.D.	0.0	1.1 × 10^−6^
Dairy products	S ^g^	milk	Module-2	Korea	N.D.	0.0	3.0 × 10^−5^
	T ^g^	milk	Module-2	Korea	N.D.	0.0	3.0 × 10^−5^
	U ^g^	milk	Module-2	Korea	N.D.	0.0	2.8 × 10^−7^
	V ^g^	milk	Module-2	Korea	N.D.	0.0	3.0 × 10^−5^
	W ^h^	yogurt	Module-2	Korea	N.D.	0.0	1.1 × 10^−7^
Meat	X ^i^	beef	Module-2	Australia	N.D.	0.0	9.9 × 10^−6^
	Y ^i^	beef	Module-2	New Zealand	N.D.	0.0	2.3 × 10^−8^
	Z ^i^	beef	Module-2	New Zealand	0.1	2.5 × 10^−5^	2.5 × 10^−5^

^a^: Several types of products. ^b^: Sample ID. ^c^: Producing district. ^d^: Dried bracken fern. ^e^: Boiled bracken fern. ^f^: Processed product. ^g^: Free-range milk. ^h^: Free-range yogurt. ^i^: Free-range beef. ^j^: Estimated exposure to PTA, where the not-detected (N.D.) results are indicated by number zero. ^k^: Estimated exposure to PTA, with the not-detected (N.D.) results indicated by the limit of detection.

## Data Availability

Not applicable.

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
