# Peer review of "Evaluation and Monitoring of the Natural Toxin Ptaquiloside in Bracken Fern, Meat, and Dairy Products"

_toxins, 2023, doi:10.3390/toxins15030231_

Round 1
Reviewer 1 Report
Review report of TOXINS-2223897
Interesting and relevant topic of the food safety for bracken fern materials in the diet. The novelty is in the many samples and the sample origin hold all the relevance of the research. Hence, the origin of sample B and Z calls for more details in the discussion. Why are those 2 samples different from the rest? Sampling place, time of the year, age of fern (B), feeding history (Z), and storage of samples prior to the buy in the shop, etc. – all crucial for the interpretation of the relevance of the manuscript.
The language is good and clear. The figures and tables are nice and clear, with some adjustments needed as indicated below.
The big question is what compound is detected by the LC-MRM method presented. In line 254, it says PTA purchased from Sigma-Aldrich but this does not seem to be available in the Sigma-Aldrich catalogue for sale. Please state CAS number to avoid mistakes for readers ordering, and alternative names used for purchase (if any) in the manuscript. The mass spectrum in Figure 1, as different from mass spectra published for ptaquiloside and several LC-MSMS methods for plant, milk and animal tissue has been published the resent 10 years, including a large range of spectra for various mass spectrometer settings in public mass spec database. First of all the sodium adduct is almost impossible to avoid even with ammonium in the eluent, hence the complete absence of sodium adducts in the mass spectrum is remarkable, e.g. 421 and 219 m/z that most often dominate the spectrum. The intense 181 fragment ought to go along the 201 fragment (201 is PTA-glucose). Also the complete missing of the degradation product pterosin B seem extraordinary (this is usually very hard to avoid in sample extraction and sample clean-up of PTA).
A thorough discussion and comparison with the LC-MRM methods in publications from the resent decade should be added before publication. The novelty for a research journal like Toxins, all rest on the improvements of the method compared to existing analytical methods and tacking the samples to illustrate the relative importance of the two sources that contained PTA above the detection limit.
Specific comments.
Line 4+ abstract short and concise
Line 16 the manuscript does not contain data that support this very definite conclusion based on ND (not detected) values except 2 single samples out of many samples.
L56-58 plenty of Na added to the samples, hence hard to believe no Na-adducts appear in the mass spectra.
L87 add what solvent is used for the standard solution.
L88 add mass range scanned to ensure 421 and 451 m/z are well included in the scan before concluding the M+H+ dominates the ESI ionisation
L254 purchase information (e.g. CAS no) and purity verification of PTA needs more details added.
L266 Shelf stability of PTA solutions is poor, and the documentation for stability during the current research to be stated to support.
L268 is this correct … diluted to …. 10, 1, 0.1 g/mL (= 10 kg/L, etc).
L280 explain d-SPE in the text (used later just as d-SPE)
L285 add abbreviation GCB as it later appears just as GCB.
L287 add abbreviation C18EC as it later appears just as C18EC.
L290 how much dilution by formic acid in water (10 times dilution?) – this influence the concentration of organic solvent in the sample when loaded to the LC, and hence the retention in the column.
L293 see note for Figure 3
L373 Reference list layout fine. Missing all international literature on analytical methods for ptaquiloside and other bracken fern toxins published the resent decade.
Also several studies of ptaquiloside, caudatoside, pterosins in milk, serum, and bracken plant tissue has been published the past decade.
Figure 1. Nice figure. I would like the x-axis to continue higher to ensure the additional adducts with higher m/z are seen, at least to 425 to include the Na-adduct of ptaquiloside.
Figure 2. Can move to Supplementary. If kept in the text, then expand and include all peaks in the chromatogram and e.g. add the ion trace for pterosin B for documentation how much (if any) has been degraded in the extraction and QuEChERS procedure (ptaquiloside is impossible to avoid some degree of transformation).
In case of ion-pairing, the peak shifts retention time, and this ought to be visible if had happened.
Figure 3. Nice figure. Add in the legend the explanation for PSA, GCB and C18EC as this is not standard abbreviations.
Figure S1. Nice figure
Figure S2. Nice figure.
Figure S3. Nice figure, but not clear from the Figure legend if the 4 replicates is all from the spiked matrix or 4 times LC-MSMS analysis of each sample. Error bars are assumed to be triplicate extractions of each of the four concentrations. Does that mean 12 extractions in total of each spike level?
Table 1. Valuable data for the manuscript.
Table 2. Why are upper bound not the same for all ND samples? Add the factor that makes the variation in f calculated upper bound for each sample.
Table S1. Nice. The monoisotopic mass of ptaquiloside is not correct.
398.44 is the molar mass and the monoisotopic mass is 398.1
Table S2. Nice layout, but the standard curve data points ought to be even-distance distributed along the concentration range tested. The current data distribution is almost entirely controlled by the response of the highest concentration.
Table S3. Nice, but is it actually correct with 5 significant figures in the regression line?
The intercept: 1079 / 37.7 = 28 (unit ? µg/kg) does not correlate with LOD 0.03 µg/kg. are the units the same in the Table, or shifted x1000 from equation to LOD?
Table S4. Nice layout. What is the difference in slope of regression lines (e.g. 9291x) compared to Table S3 (37.7x)?
The intercept in Table S4 is all below 1 compared to 28 in Table S3. Not clear how to read the Standard solution regression line.
Reviewer 2 Report
This paper details Evaluation and Monitoring of the Natural Toxin Ptaquiloside in Bracken Fern, Meat, and Dairy Products.
It is an interesting and complete study, with high applicability, a lot of experimental methods being used.
This paper can be published in Toxins.
Reviewer 3 Report
The manuscript is acceptable in actual form.
Reviewer 4 Report
The authors have presented an interesting manuscript ˝Evaluation and Monitoring of the Natural Toxin Ptaquiloside in Bracken Fern, Meat, and Dairy Products˝ still some parts of the manuscript need to be improved.
1. I suggest the authors add in the Supplementary material Total Ion Chromatogram Full Scan of one of the samples. It is interesting to the scientists that work in the same field.
2. Section 2.1.2. - only the list of the optimized parameters with the corresponding range is given. The authors need to give some explanation on the selection of the most suitable ones. The influence of each parameter on method performance is an important part of analytical method optimization.
3. The valuable information for the scientists that work in the same field is the mass spectrum of analyte in standard solution as well as in the sample.
4. According to ICH guidelines stability of the analyte in the standard solution and in the sample needs to be evaluated.
5. Table S2. - please correct of formic acid (twice)
Reviewer 5 Report
The paper entitled “Evaluation and Monitoring of the Natural Toxin Ptaquiloside in Bracken Fern, Meat, and Dairy Products” is nicely written and well organized. The results are well described and clearly presented. The paper should be accepted for publication. I would just suggest the authors to add a paragraph in Introduction section about other methods that can be used for the same purpose.
Round 2
Reviewer 1 Report
Thanks for the revised manuscript and the additions of full-scan mass spectrum and additional details for methods used. These improvements are valuable and underline the quality of the experimental data provide in the manuscript.
One question: in line 100 you use 1 g/mL PTA for the infusion and this is usually 1 mg/L for MS (1 million times lower). Please consider if this is a typo, and the m for milli should be moved to above the / sign.
Otherwise I happy with the manuscript in its current form.